# Investigating How Genomic Contexts Impact IS5 Transposition Within the *Escherichia coli* Genome

**DOI:** 10.3390/microorganisms12122600

**Published:** 2024-12-16

**Authors:** Jonathan Onstead, Zhongge Zhang, Jialu Huo, Jack W. Ord, Sofia Smith, Milton H. Saier

**Affiliations:** Department of Molecular Biology, School of Biological Sciences, University of California at San Diego, 9500 Gilman Dr, La Jolla, CA 92093-0116, USA; jonstead@ucsd.edu (J.O.); j1huo@ucsd.edu (J.H.); jord@ucsd.edu (J.W.O.); ssmith@ucsd.edu (S.S.)

**Keywords:** IS5, transposons, transposable elements, genomic context, Ins5A, Ins5B, Ins5C, transposition mechanism, cis effect

## Abstract

Insertions of the transposable element IS5 into its target sites in response to stressful environmental conditions, DNA structures, and DNA-binding proteins are well studied, but how the genomic contexts near IS5′s native loci impact its transpositions is largely unknown. Here, by examining the roles of all 11 copies of IS5 within the genome of *E. coli* strain BW25113 in transposition, we reveal that the most significant copy of IS5 is one nested within and oriented in the same direction as the *nmpC* gene, while two other copies of IS5 harboring point mutations are hardly transposed. Transposition activity is heavily reliant on the upstream *nmpC* promoter that drives IS5 transposase gene *ins5A*, with more transpositions resulting from greater promoter activity. The IS5 element at *nmpC* but not at other loci transcribed detectable amounts of *ins5A* mRNA. By increasing expression of the *ins5CB* operon harbored in IS5, we demonstrate that Ins5B and Ins5C appear to exert a stimulatory role in IS5 transposition, suggesting that the downstream genomic regions near the native loci are involved in overall IS5 transposition as well. Using a strain that carries IS5 only at the *nmpC* locus, we confirm that IS5 primarily uses a copy/paste mechanism for transposition, although we cannot rule out the cut/paste mechanism.

## 1. Introduction

Insertion sequences (ISs) are the simplest transposable DNA elements that are capable of transposition from one location to another within a prokaryotic genome or from a plasmid to the chromosome (or vice versa) in bacterial cells [1]. As the smallest transposons (usually 0.7 kb to 2 kb in length), IS elements are ubiquitous and widely distributed across bacterial genomes, often with multiple distinct elements per genome [2,3]. These small transposons have been shown to play important evolutionary impacts on the structure and function of prokaryotic genomes [1,4]. Several major genetic effects have been documented, involving genome rearrangements [5,6,7,8], partial deletions [9,10,11], and the activation/inactivation of neighboring genes [12,13]. As important players in shaping prokaryotic genomes, these insertion sequences are now recognized as a (variable) constituent of their host genomes [1,4,14]. On the other hand, these mobile elements have been recognized as one main causal agent promoting multidrug resistance in bacterial pathogens [15,16,17].

Originally discovered from the AT-rich immunity region of coliphage lambda [18], IS5 is a well-studied IS element commonly distributed in natural isolates of *E. coli* [19]. As shown in Figure 1, IS5 is 1195 bp in length and is flanked by two 16 bp imperfect inverted repeat (IR) sequences [20,21]. As a member of the IS5 family, it encodes a “DDE”-type transposase Ins5A [14], vital for independent transposition. In addition, IS5 carries a second operon (*ins5CB*) within the *ins5A* gene, which is transcribed in the opposite direction and encodes two small proteins of unknown function [22]. This element harbors a sharply curved DNA structure located at the 3′ region, which is AT-rich and contains a series of A-tracts with a cortically localized IHF binding site [23]. Once inserted, it usually causes a direct repeat of the 4 bp target site (with the consensus sequence CA/TAG/A) flanking the element [20,21]. In one case, IS5 has been shown to precisely excise from its location on the *E. coli* chromosome [24].

IS5 can hop to multiple sites on single chromosomes, and its copy number varies widely, ranging from 0 to 23 per genome among different *E. coli* strains [19,25]. *E. coli* K12 strain BW25113 carries 11 copies of IS5 located at 11 well-separated loci on its chromosome [26]. Like other IS elements, IS5 insertion leads to chromosomal rearrangements [25] and gene disruption or inactivation [27,28,29,30]. However, most studies have focused on IS5-mediated elevation or activation of gene expression [31,32,33,34,35,36]. IS5 insertion relies on local DNA structural features, being biased towards target sites within intergenic regions, AT-rich regions, the specific SIDD (superhelix stress-induced DNA duplex destabilization) sequences, and H-NS-bound DNA regions [37,38,39].

During the past decade, it has been established that IS5 transposition usually occurs at greater frequencies in response to stressful environmental conditions. Such a transpositional event is regulated by host proteins, especially nucleoid-associated proteins (NAPs) and transcriptional regulatory proteins [40]. IS5 insertion into the regulatory region of the *glpFK* operon, leading a Glp^+^ phenotype in a *crp* deletion genetic background, is promoted by the presence of glycerol but was prevented by the presence of GlpR and Crp, which bind to a downstream region near the IS5 insertion site [36]. IS5 insertion upstream of the cryptic *bglGFB* operon, leading to a positive β-glucoside growth (Bgl^+^) phenotype, preferentially occurs in the presence of a β-glucoside such as arbutin or salicin. This insertional process involves the operon anti-terminator BglG, which likely acts to stabilize the SIDD structure, thus facilitating IS5 insertion [41]. Another example involves IS5 insertion into the *flhDC* regulatory region, which occurs preferentially in *E. coli* cells growing within soft agar but not on hard agar or in liquid media [42,43]. Lastly, IS5 usually transposes into a single site upstream of the *fucAO* operon, leading to a positive propanediol growth (PPD^+^) phenotype [24,34]. IHF, a nucleoid-associated protein, has been shown to play a crucial role in all these activating mutations, probably by altering the DNA conformation of IS5 in its native locations [40]. In all these cases, the IS5 insertion site is buried within a SIDD sequence [38], which may be altered upon binding by local or global DNA-binding proteins.

For a successful transposition event, two key genomic regions are directly involved, namely the recipient site (to receive the incoming IS element) at one chromosomal location and the donor site (to donate the transposing IS element) at another location. Thus far, most of the investigations concerning IS transpositions have been concentrated on how IS elements insert into their recipient sites (target sites or hotspots), such as their dependence on environmental conditions and favorable DNA structures, and their regulation by DNA-binding proteins. Efficient transpositions depend not only on an ideal hotspot (including its surrounding regions) but also on the donor locus (IS element’s native site) for the transposing IS element and its adjacent genomic contexts. However, this latter genomic factor has been overlooked in terms of its role in overall IS transposition. Thus, little information is available concerning how the genomic contexts flanking the transposing elements impact transposition frequencies. This study aims to fill in this gap, focusing on the IS5 element.

As mentioned above, there are 11 copies of IS5 distributed on the chromosome of strain BW25113, each having a distinct genomic region. It has been reported that the endogenous promoters for the transposase gene *ins5A* and the *ins5CB* operon are largely inactive, with little transcription detected [40,44]. It is unknown how these IS5 elements contribute to overall transposition and how the genomic DNA factors affect the expression of *ins5A* and *ins5CB*. If some IS5 copies contribute to the total transposition more than the others, it would be of interest to examine how the specific genomic contexts impact IS5 transposition frequency. In addition, little attention has been given to the two small proteins, Ins5C and Ins5B, encoded on IS5. Here, we show that the IS5 element embedded within the *nmpC* gene plays the most important role in overall IS5 transposition, while two mutant elements at *lomR* and *cobU*/*yoeG* loci are essentially incapable of transposition. The transposition activity depends on the promoter upstream of the *nmpC* being proportional to the promoter strength. Among 11 copies, only the IS5 element at *nmpC* yields a detectable level of *ins5A* mRNA. The overexpression of *ins5CB* at the *nmpC* locus substantially promotes IS5 transposition, suggesting that these two small proteins, Ins5B and Ins5C, play a role in IS5 transposition. Lastly, we confirm that IS5 primarily employs a “Copy/Paste” mechanism for transposition.

## 2. Materials and Methods

### 2.1. E. coli Strains and Growth Conditions

*E. coli* K12 strain BW25113 [45] was used as the wildtype strain. All other strains used in this study were derived from this strain, and they are described in Appendix A.

Bacterial strains were routinely cultured in LB media at 30 °C or 37 °C. For the β-glucoside growth (Bgl^+^) mutation assay, a propanediol growth (PPD^+^) mutation assay or 5′ RACE experiment with M9 media with an appropriate carbon source were used. The 10× M9 salt solution (per liter) contained 60 g of Na_2_HPO_4_, 30 g of KH_2_PO_4_, 10 g of NH_4_Cl, and 5 g of NaCl. After diluting to 1× M9 medium, it was supplemented with 1 mM MgSO_4_ and 0.1 mM CaCl_2_. When necessary, ampicillin (Ap), kanamycin (Km), and chloramphenicol (Cm) were added to the media at 100 μg/mL, 25 μg/mL, and 10 μg/mL, respectively.

### 2.2. Construction of IS5 Deletion Mutants

Using the Lambda Red approach, nine IS5 elements with no point mutations were individually deleted from the chromosome of strain BW25113. For those copies of IS5, flanked by two insertion sites such as CTAA or CTAG, the downstream site was deleted together with the IS5 element. These nine IS5 copies are distributed across the chromosome at the loci of *ykfC*, *nmpC*, *gltI*/*lnt*, *ynaI*/*ynaJ*, *wbbL*, *yejO*, *yghO*, *yhcE*, and *yhiS*, respectively. The kanamycin (Km) resistance gene (*km^r^*) was amplified from pKD13 using a pair of chimeric oligos (Appendix A). The PCR products were gel purified and then electroporated into BW25113 cells (expressing Lambda Red proteins encoded by pKD46) to substitute for the IS5 element of interest. For each IS5 mutation, several Km-resistant colonies were subject to PCR verification for the replacement of IS5 by the *km^r^* gene. The *km^r^* gene was flipped out by pCP20. This yielded nine single IS5 deletion mutants that were named ZZ245, ZZ246, ZZ248, ZZ249, ZZ250, ZZ251, ZZ252, ZZ253, and ZZ254, respectively (Appendix A). Similarly, the *nmpC* gene was replaced by the *km^r^* gene that was subsequently flipped out. This yielded strain ZZ247, in which both *nmpC* and the IS5 element buried within *nmpC* were deleted.

At the loci of *nmpC*, *wbbL, yejO*, and *gltI/lnt*, IS5 is oriented in the same direction as the target gene. Using three rounds of P1 transduction followed by *km^r^* gene removal, these 4 copies of IS5 were deleted all together in BW25113, yielding a quadruple mutant ZZ255. To see if the IS5 element alone can transpose, another eight copies of IS5 were all deleted using the same process (i.e., multiple rounds of P1 transduction followed by *km^r^* flipping out), yielding strain ZZ263 that deleted all active IS5s except the IS5 element within the *nmpC* gene.

### 2.3. Deletion or Blockage of the nmpC Promoter (P_nmpC_)

At the *nmpC* locus, the IS5′s transposase gene, *ins5A*, is transcribed in the same direction as *nmpC*. The *nmpC* promoter P*_nmpC_* is likely to transcribe through *ins5A* at this locus, leading to IS5 transposase production. To explore P*_nmpC_*’s impact on overall IS5 transposition, this promoter was either deleted or blocked. To delete P*_nmpC_*, the 199 bp upstream region at positions −199 to −1 with respect to the *nmpC*’s start codon was replaced by a *km^r^* gene amplified from pKD13. The *km^r^* gene was subsequently flipped out by pCP20, yielding strain ZZ256 that was deleted for P*_nmpC_*. To block P*_nmpC_*, an *rrnB* terminator together with a *km^r^* marker (*km^r^*:*rrnB*T) were amplified from pKDT [46] and were then inserted immediately downstream of the *nmpC*’s start codon in strain BW25113. After the *km^r^* gene was flipped out, this yielded strain ZZ257, in which the P*_nmpC_* and any upstream transcription were blocked by the inserted terminator.

### 2.4. Construction of P_tet_ Substitution for P_nmpC_ on the Chromosome

Using plasmid pKDT:P*_tet_* [46] as the template, the DNA cassette (referred to as “*km^r^*:*rrnB*T:P*_tet_*”) containing the *km^r^* gene, the *rrnB* terminator (*rrnB*T), and the P*_tet_* promoter was amplified using the primer pair Ptet-nmp-P1/Ptet-nmp-P2 (Appendix A). The PCR products were integrated into the chromosome of strain BW25113 to replace P*_nmpC_* (−210 to the −1 relative to the translational start point of *nmpC*). The chromosomal integration was confirmed first by colony PCR, followed by DNA sequencing. This yielded strain ZZ258, in which P*_tet_* drives *nmpC* and the *ins5A* gene nested in *nmpC*, while P*_nmpC_* is deleted. The region carrying “*km^r^*:*rrnB*T:P*_tet_*-*nmpC*” was transferred to strain BW-RI (constitutively expressing TetR) [47] by P1 transduction, yielding strain ZZ259. In this resultant strain, the P*_tet_* activity was inversely proportional to the levels of TetR, which was released from P*_tet_* upon addition of anhydro-tetracycline (aTc). In other words, the more aTc added, the greater the P*_tet_* promoter activity.

### 2.5. Insertion of a Terminator or P_tet_ Downstream of nmpC

In addition to *ins5A*, IS5 carries two other genes, *ins5C* and *ins5B*, which form an *ins5CB* operon with its transcription overlapping but opposite to *ins5A*. Little information is available concerning the function(s) of these two small proteins. To determine the possible effect of downstream genomic regions on IS5 transposition, an *rrnB* terminator was added between *nmpC* and *euuD*, replacing the small region located at −16 to −41 with respect to the stop codon of *euuD*. This yielded strain ZZ260, in which any possible transcription from the *nmpC*’s downstream region (potentially transcribing through *ins5CB*) was blocked. To further examine the possible role of Ins5C and Ins5B, a P*_tet_* promoter was inserted downstream of *euuD*, replacing the region between position +9 to +78 relative to the stop codon of *nmpC*. In the resultant strain, ZZ261, the added P*_tet_* promoter, should transcribe the *ins5CB* operon.

### 2.6. Construction of P_tet_ Driving ins5A at the intS Locus

The “*km^r^*:*rrnB*T:P*_tet_*” cassette, containing a *km^r^* gene, an *rrnB* terminator (*rrnB*T), and the P*_tet_* promoter, was amplified from pKDT:P*_tet_* [46] using oligos intS-km-P1 and ins5A-Ptet-R (Appendix A). The *ins5A* structure gene was amplified from BW25113 genomic DNA using Ptet-ins5A-F and ins5A-int-P2. These two fragments (*km^r^*:*rrnB*T:P*_tet_* and *ins5A*) were combined by fusion PCR using oligos intS-km-P1 and ins5A-int-P2. The fusion product, *km^r^*:*rrnB*T:P*_tet_*-*ins5A*, was integrated on the chromosome to replace *int*S (−48 to +1114 relative to the *intS* translational initiation site). This yielded strain ZZ264 that harbors P*_tet_*-driven *ins5A* at the *intS* locus.

### 2.7. β-Glucoside Growth Mutation (Bgl^+^) Assay

Bgl^+^ mutation assays were performed as previously reported [41] on minimal M9 agar plates with 0.5% of salicin (a β-glucoside) as the sole carbon source. Briefly, a fresh colony from each test strain was cultured in LB liquid medium with shaking for about 8 h at 30 °C, washed twice using 1× M9 salts (carbon source-free), and applied onto agar plates (2 × 10^7^ cells/plate). The plates were then incubated in a 30 °C incubator and were examined daily for the appearance of Bgl^+^ colonies, with each colony representing a new Bgl^+^ mutation. On these β-glucoside minimal agar plates, any colonies appearing by day 2 after plating were considered to be from Bgl^+^ cells initially applied onto the plates. They were therefore subtracted from the subsequent measurements.

The total numbers of Bgl^−^ cells (background populations) were determined as described previously [36,41,48]. The frequencies of Bgl^+^ mutations on salicin M9 plates were determined by dividing the total Bgl^+^ colonies by the number of Bgl^−^ colonies. The mutation frequencies were normalized to Bgl^+^ mutations per 10^8^ cells at a given period of time.

### 2.8. Swarming Mutation Assay

The swarming mutation assay was performed following the approach of Barker et al. [35]. A fresh colony was inoculated in 3 mL LB medium within a glass tube, and the tube was shaken (200 rpm) overnight at 30 °C. The culture was washed once with M9 salts and diluted to an OD_600_ of 1.0 prior to use. A total of 1 µL of the cell suspension was inoculated inside the LB semisolid agar (0.3%) within a 9 cm plate with three inoculations per plate. Four plates were used for each test strain. The plates were incubated at 30 °C. The swarming mutants, represented by outgrowths of motile subpopulations from the colonies, were counted after 20 h and 25 h of incubation, respectively.

### 2.9. Propanediol Growth (PPD^+^) Mutation Assay

For the PPD+ mutation assay, cell suspensions were prepared as for the Bgl^+^ mutation assay. About 100 million cells were applied onto each of the agar plates with 1% propanediol as the sole carbon source. The plates were incubated at 30 °C prior to daily examination of the PPD^+^ mutants. The frequencies of PPD^+^ mutations were determined by dividing the total PPD^+^ colonies by the number of PPD^−^ cell populations applied.

### 2.10. TSS Determination Using a SMARTer^®^ RACE 5′/3′ Kit

As mentioned above, there are 11 copies of IS5 on the chromosome of wildtype strain BW25113. These IS5 copies are expected to synthesize various amounts of transposases due to their orientations on the chromosome and the different upstream genomic regions. To see which copies of IS5 can yield detectable *ins5A* mRNA, we decided to identify the transcriptional start site(s) (TSS) of the target genes using an *ins5A* specific oligo. Strain BW25113 was shaken at 37 °C in M9 minimal media with 0.5% glycerol as the carbon source. RNase inactivation, total RNA preparation, rRNA removal, and mRNA extraction were carried out as reported by Zhang et al. [49].

Using the SMARTer^®^ RACE 5′/3′ kit (Takara Bio, San Jose, CA, USA), first-strand cDNA was synthesized following the method recommended by the supplier. The extracted mRNA sample was first combined with a random hexamer mixture and was then incubated at 72 °C for 3 min followed by 42 °C for 2 min. A buffer containing RNase inhibitor, Reverse Transcriptase, and SMARTer II Oligonucleotide (all provided) was added to the mRNA/random hexamer mixture sample. After incubation at 42 °C for 90 min and then 70 °C for 10 min, the resulting mixture (first-strand cDNA) was diluted with tricine–EDTA buffer. After dilution, the first-strand cDNA was combined with a PCR master mix, the primer IS5-GSP-R (binding to *ins5A* near the start codon), and the universal primer mix for amplification. The amplified products (amplified cDNA) were gel purified and then subject to DNA sequencing analysis using IS5-GSP-R (and one or two other primers specific to target genes if necessary). The first nucleotide immediately downstream of the SMARTer II Oligonucleotide sequence (adaptor) was the transcriptional start site (+1) of the target gene.

### 2.11. Statistical Analysis

All IS5 insertion frequency data are expressed as the mean ± standard deviation (SD). Statistical significance was tested by either a two-sample *t*-test (for 2 treatments) or a 1-way ANOVA followed by Tukey–Kramer’s post hoc test (for ≥ 3 treatments) using RStudio (Version 2023.12.0 + 369 “Ocean Storm” Release for Windows). All figures and mutation frequencies were generated using Microsoft Excel (Version 16.66.1). Details of the statistical tests used are indicated in the figure legends. NS denotes no significance and indicates a *p*-value of ≥ 0.05; * indicates a *p*-value < 0.05; ** indicates a *p*-value < 0.01; *** indicates a *p*-value < 0.001; and **** indicates a *p*-value < 0.0001.

## 3. Results

### 3.1. Two Copies of IS5 with Point Mutations Are Incapable of Transposition

Table 1 lists all eleven IS5 elements distributed on the chromosome of strain BW23113, providing their chromosomal locations, target sites, orientations (relative to the target genes), and similarity to the wildtype IS5. These IS5 elements are located across the chromosome with the shortest distance being 30 kb between the *ynaI*/*yna*J and *lomR* loci. Four copies of IS5, located within or near the *nmpC*, *lnt*, *wbbL* and *yej*O genes, respectively, are oriented with their transposase genes transcribed in the same direction as their target genes. Among the 11 copies of IS5, two of them carry point mutations. Located at the *lomR* locus, the IS5 element shows only 91.5% similarity to the wildtype element.

Single bp insertions, two single bp deletions, and 96 mismatches were found (Appendix A). These mutations lead to a two-residue shorter transposase with six altered residues (Appendix A). For IS5 at the *cobU*/*yoeG* site, there are five mismatches, leading to an R74K mutation in the transposase Ins5A (Appendix A). The point mutations present within these two mutant IS5 copies were confirmed by DNA sequencing.

To see if these mutant IS5 elements can transpose, we first conducted three mutation assays using strain BW25113, including β-glucoside growth (Bgl^+^) mutations, propanediol growth (PPD^+^) mutations, and swarming mutations (SWM^+^). From each type of mutation assay, twenty IS5 insertional mutants (identified by colony PCR) were chosen for further analyses. The regulatory regions of the *bglGFB* operon (Bgl^+^ mutants), the *fucAO* operon (PPD^+^ mutants), and the *flhDC* operon (SWM^+^ mutants) were amplified and subsequently subject to DNA sequencing analyses. The nucleotide sequences of the IS5 elements present in the amplified products were examined for the presence of point mutations observed for those two mutant IS5 elements located at the *lomR* and *cobU*/*yoeG* loci. Among the 60 independent mutants isolated from three types of mutation assays, no point mutations were found with each of the inserted IS5 elements. These results indicate that these two IS5 elements carrying multiple mutations are either incapable of transposition or they might transpose at extremely low frequencies (below a detectable level).

### 3.2. IS5 at nmpC Plays a Major Role in IS5 Insertion Upstream of the Bgl Operon

Except those two mutant IS5 elements (unable to transpose) described above, the remaining nine IS5 elements were individually deleted from BW25113. Using Bgl^+^ mutation assays, these deletion mutants were compared with the wildtype strain for IS5 transposition into the regulatory region of the *bglGFB* operon. The results are summarized in Figure 2. Figure 2A shows the frequencies of Bgl^+^ mutations that occurred on day 2 post-plating (note that colonies were counted on day 4 post-plating). Among these elements, the deletion of IS5 at *nmpC* (strain ZZ246) yielded a fourfold reduction in IS5 insertion frequency, while the deletion of other copies either did not affect IS5 insertions or affected the insertions to much lesser degrees. Similar observations were made for day 6’s Bgl^+^ mutations (Figure 2B), with a >3-fold decreased mutation frequency observed for the strain deleted for IS5 from *nmpC*. Deletion of the full *nmpC* gene together with the nested IS5 element had virtually the same effect as the IS5 element deletion alone (Figure 2A,B). In addition, a moderate decrease in the insertion frequency was observed when the IS5 element was deleted from the *lnt* or *wbbL* locus.

Like IS5 at *nmpC*, three other copies of IS5, located at *lnt*, *wbbL*, and *yejO*, respectively, lie in the same directions as their respective target genes (Figure 3A). At the *lnt* locus, in addition to the *lnt* promoter, there are three other putative promoters with unknown function, P*_insH3_*, P*_insH5_*, and P*_insH6_*, located in the *lnt*/IS5 intergenic region [50]. At the *wbbL* locus, in addition to the *wbbL* promoter (P*_wbbL_*), three other internal putative promoters (P*_insH4_*, P*_insH5_*, and P*_insH6_*) are present upstream of the IS5 element as well [50]. At the *yejO* locus, only a putative *yejO* promoter (P*_yejO_*) is located upstream of the IS5 element. If these upstream promoters are true and active, they are likely to promote IS5 transposition, as they may drive the transcription of the respective *ins5A* gene.

To see if there is a combined or synergistic effect among these IS5 elements, these four copies of IS5 were simultaneously deleted from BW25113, yielding a quadruple mutant ZZ255. This quadruple mutant as well as four single mutants were subject to Bgl^+^ mutation assays over time, and the results are summarized in Figure 3B. Among the four single mutants, the lack of IS5 at the *yejO* locus did not appreciably affect the Bgl^+^ mutation frequency over time. A moderate (roughly 30%) reduction in mutation frequency was observed when th*e lnt* copy or the *wbbL* copy of IS5 was removed. The largest changes, that is, three- to five-fold fewer mutations, were again seen when the *nmpC* copy of IS5 was deleted. However, when all four copies of IS5 were removed, no further decrease in mutation frequency was observed as compared to the single deletion of IS5 at *nmpC*. Based these results, we conclude that (i) the IS5 element embedded within *nmpC* contributes the most to overall IS5 transposition; (ii) IS5 elements at the *lnt* and *wbbL* loci play a moderate role in IS5 transposition; (iii) all other copies of IS5 do not substantially affect IS transposition; and (iv) no additive or synergistic effect is shown among these four copies of IS5. Considering these observations, we chose to focus on characterizing the *nmpC* copy of IS5 in the further sections of this paper.

### 3.3. IS5 at nmpC is Important for Other IS5-Targeted Gene Mutations

Using the Bgl^+^ mutation assays, we demonstrated that the IS5 element at the *nmpC* locus is the most important transposing element among the 11 copies of IS5 in the BW25113 genome. Next, we directed our investigation towards other IS5 insertional mutations, including swarming mutations (SWM^+^) upstream of the *flhDC* operon and propanediol growth (PPD^+^) mutations upstream of the *fucAO* operon. Figure 4 summarizes the mutation frequency results obtained using both assays.

The SWM^+^ assays were performed in LB soft agar plates. After a 20 h incubation, the IS5 deletion cells (ZZ246) displayed 2.5-fold fewer IS5 insertional SWM^+^ mutations than the wildtype cells (left panel of Figure 4A). Similarly, after a 25 h incubation, the lack of IS5 at *nmpC* led to a nearly twofold decrease in SWM^+^ mutations (right panel of Figure 4A). The PPD^+^ mutations were conducted on minimal agar plates with propanediol as the sole carbon source. After both 4-day and 8-day incubations, there was a threefold difference in mutation frequencies between the IS5-lacking strain (ZZ246) and the wildtype strain, with the former exhibiting fewer mutations (Figure 4B). To summarize, the results from all three mutation assays (Bgl^+^, SWM^+^, and PPD^+^) clearly indicate that the lack of the *nmpC* copy of IS5 leads to a significant decrease in IS5 transposition activity. Based on these results, we conclude that IS5 located at *nmpC* is the primary contributor for an overall increase in IS5 transposition frequency. This observation is surprising given that there are eight other active copies of IS5 within the test strain.

### 3.4. nmpC Promoter Is Critical for the Overall IS5 Transposition Rate

All nine active copies of IS5 are identical in their nucleotide sequences and have no point mutations present within these full elements. However, each IS5 element is flanked by distinct genomic DNA sequences. The *nmpC* copy of IS5 dictated the overall transposition frequency, likely due to the unique chromosomal sequences surrounding the element. There is a typical σ70 promoter (P*_nmpC_*) upstream of *nmpC*, which conceivably drives transcription not only of the interrupted *nmpC* gene but also of the transposase gene (*ins5A*), since they are oriented in the same direction (see “wildtype” in Figure 5A). Strain ZZ246 carries the deletion of IS5 at *nmpC* with an 85 bp FRT scar remaining. To examine the role of P*_nmpC_*, strains ZZ256 and ZZ260 were made (Figure 5A), which either carry a deleted P*_nmpC_* (ZZ256) or an inserted *rrnB* terminator immediately downstream of P*_nmpC_* (ZZ260). In the latter case (strain ZZ260), the activity of P*_nmpC_* was blocked by the strong terminator.

These strains were assayed for standard Bgl^+^ mutations using salicin minimal agar plates, and the results were listed in Figure 5B. As can be seen, the deletion of P*_nmpC_* (strain ZZ256) and the blockage of P*_nmpC_* (ZZ260) led to similar mutation frequencies, which were roughly 2.5-fold lower than the wildtype strain. These results demonstrate that P*_nmpC_* is required for the high overall IS5 transposition activity, suggesting that P*_nmpC_* indeed drives *ins5A* transcription and that the increased rate of transcription is required for normal IS5 transposition. On the other hand, the lack of P*_nmpC_* had a mildly lesser effect than the lack of the entire IS5 element at *nmpC* on IS5 insertional Bgl^+^ mutations (that is, a 2.5-fold reduction for DP*_nmpC_* vs. a 3.5-fold reduction for DIS5). The main reason for this observation could be that the *nmpC* copy of IS5 itself within the strain deleted for DP*_nmpC_* or the strain with a terminator blocking P*_nmpC_* still maintains some transposition activity when P*_nmpC_* is absent or is blocked. Nevertheless, the key reason for the IS5 element at *nmpC* to be important in transposition is because of the presence of P*_nmpC_*, which transcribes the transposase gene, thereby increasing IS5 transcription activity.

### 3.5. IS5 at the nmpC Locus is the Only Element That Efficiently Transcribes ins5A

Since the *ins5A* native promoter is extremely weak, most *ins5A* transcription is expected to be derived from the genomic regions upstream of inserted IS5 elements, especially the regulatory regions of target genes. In this case, *ins5A* and the related target gene would be transcribed simultaneously, forming a fusion mRNA with the transcriptional start site (TSS) being the same as the target gene’s TSS. Total mRNA samples plus all target gene/IS5 fusion mRNAs were converted to the cDNAs that were used for identifying the target genes by DNA sequencing. To examine which copies of IS5 yielded detectable *ins5A* mRNA, we employed 5′ RACE to determine the TSSs using the reverse IS5-specific oligo (IS5-GST-R) (Appendix A) that binds to *ins5A* about 200 bp downstream of its start codon. If the fusion mRNA is relatively long, another specific reverse oligo bound to the target gene/operon near the start codon would be needed to reach the TSS. Using this approach, the target genes which are transcribed with *ins5A* can be identified by DNA sequencing.

Wildtype strain BW25113 was cultured with M9 glycerol minimal medium prior to mRNA preparation and purification, cDNA synthesis, and amplification. Amplified cDNA samples were subject to agarose gel electrophoresis, and two cDNA product bands (0.25 kb and 1.4 kb in length) were detected, with the bigger band being brighter (Figure 6A). Except for these two bands, no other DNA bands were visible. The subsequent DNA sequencing analysis of the 0.25 kb product by the same IS5 oligo IS5-GST-R revealed the TSS to be the nucleotide “A”, which is the same TSS reported for the *ins5A* native promoter within IS5 (Figure 6B). Of note, this weak cDNA band resulted from 11 chromosomal copies of IS5 present in wildtype strain BW25113. In addition, to make the cDNA band visible, a relatively large amount of total mRNA was used for cDNA synthesis. These results are consistent with the fact that the *ins5A* resident promoter is extremely weak.

The 1.4 kb cDNA product was identified to be an *nmpC*/IS5 fusion product with the IS5 target site CTAA in between (left panel of Figure 6C). Further DNA sequencing analysis using a *nmpC*-specific reverse oligo (nmpC-GST-R) revealed the nucleotide “G” as the TSS for the *nmpC*/IS5 fusion mRNA, which was identical to the TSS for P*_nmpC_* (right panel of Figure 6C). With these results, we conclude that (i) the *nmpC* copy of IS5 efficiently transcribes the transposase gene *ins5A* with the primary transcription to be from P*_nmpC_*; (ii) other copies of IS5 together with their adjacent genomic regions do not synthesize detectable levels of *ins5A* mRNA; and (iii) the endogenous *ins5A* promoters alone, probably within all eleven IS5 elements, are able to produce a low level of *ins5A* mRNA under our experimental conditions.

Figure 6D shows the nucleotide sequences of the P*_nmpC_*, part of *nmpC*, and part of IS5 with annotations. Highlighted in green (−35 motif, −10 motif, and TSS), P*_nmpC_* serves to transcribe not only the IS5-interrupted *nmpC* gene but also the *ins5A* transposase gene. This is the only copy of IS5 that yielded a detectable level of *ins5A* mRNA. P*_nmpC_* is activated by Crp bound to its operators O*_Crp_*1 and O*_Crp_*2, while it is repressed by OmpR (bound to its operator O*_OmpR_*), IHF (bound to O*_IHF_*), and MprA (bound to O*_MprA_*). In addition, the expression of the *nmpC* gene together with its nested *ins5A* gene is under the post-transcriptional control mediated by the small RNA RyhB. Likely, these regulators would impact the transposition activity of IS5 within BW25113 by regulating the *ins5A* transcription. The native *ins5A* promoter P*_ins5A_* (−35 motif, −10 motif, and TSS) at the 5′ region of IS5 is highlighted in cyan. The joint effort from all native P*_ins5A_* promoters harbored within 11 copies of IS5 would result in a low level of *ins5A* transcription.

### 3.6. IS5 Transposition Activity Is Proportional to the Strength of the Promoter Upstream of nmpC

To titrate *ins5A* expression, the promoter P*_tet_* was first substituted for P*_nmpC_* in the wildtype strain, yielding ZZ258 (Figure 7A), in which P*_tet_* exhibited its maximal activity due to the absence of TetR (the repressor for P*_tet_*). The P*_tet_* construct was transferred to BW-RI that constitutively expresses *tetR*, yielding ZZ259 (Figure 7A). In the absence of anhydrotetracycline (aTc), P*_tet_* was off due to the TetR binding and thus excluding RNA polymerase. When aTc is present, P*_tet_* was on due to the release of TetR. Greater amounts of aTc added led to greater P*_tet_* promoter activities.

Strains ZZ258 (no TetR) and ZZ259 (with TetR), both with P*_nmpC_* being replaced by P*_tet_*, were compared with the wildtype using Bgl^+^ mutation assays. As shown in Figure 7B, strain ZZ258 (with the maximal promoter activity) displayed an even slightly higher mutation frequency than the wildtype. This could be because P*_tet_* has a greater activity than P*_nmpC_*. In the case of strain ZZ259 (with TetR), an approximately twofold decrease in Bgl^+^ mutation frequency was seen in the absence of aTc (strong repression of P*_tet_*) in comparison to the wildtype. It is of note that without aTc, P*_tet_* might not be fully shut off by TetR. When more aTc was added, there were higher P*_tet_* activities and thus more Bgl^+^ mutations. These results support the notion that IS5 transposition activity depends on the strength of the promoter driving *nmpC*, consistent with the earlier observation (Section 3.4) that P*_nmpC_* plays a critical role in increasing IS5 transposition.

### 3.7. Increased Expression of ins5A at the intS Locus Does Not Promote IS5 Transposition

To assess if the transposase has a trans effect on IS5 transposition, the promoter P*_tet_* was used to drive the transposase gene *ins5A* alone at the *intS* locus (Figure 8A), an ideal chromosomal locus for the expression of genes of interest [41,44], which is far away from all 11 copies of IS5 on the chromosome. The resultant *ins5A* overexpression strain, ZZ264, was examined for the standard Bgl^+^ mutations in comparison to the wildtype strain. As shown in Figure 8B, *ins5A* expression at the *intS* locus did not appreciably enhance the transposition of IS5 elements (located on other distal chromosomal loci) into the upstream hotspot of the *bglGFB* operon. Similar observations were made when using SWM^+^ and PPD^+^ mutation assays. These results suggest that the IS5 transposase mainly exerts a cis effect, preferentially transposing the element that encodes it.

### 3.8. Blockage of the Downstream Genomic Region Near nmpC Has Little Effect on IS5 Transposition, But Increased ins5CB Expression Enhances It

As described above, we demonstrate that the upstream genomic region of IS5 at *nmpC* is required for regular IS5 transposition. Next, we focused on examining the role of the downstream region. Opposite to *ins5A*, there is an *ins5CB* operon within IS5, whose promoter (like P*_ins5_A*) has little activity [40,44]. Downstream of the inserted IS5 element are the *quuD* gene (Figure 9A) and several other operons that are transcribed in the same direction as *ins5CB*. The promoters *for quuD* (P*_quuD_*) and other downstream operons may impact *ins5CB*. To block possible transcription from these promoters through the *ins5CB* operon, a strong *rrnB* terminator was added in the *nmpC*/*quuD* intergenic region (strain ZZ260). To see the effect of *ins5CB* overexpression on transposition, a constitutive promoter P*_tet_* was inserted almost immediately downstream of *nmpC*, directly driving *ins5CB* (strain ZZ261).

Using standard Bgl^+^ mutation assays, strains ZZ260 (with an added terminator) and ZZ261 (with P*_tet_* driving *ins5CB*) were tested for IS5 transposition. As seen in Figure 9B, the addition of the strong terminator did not appreciably affect IS5 insertional Bgl^+^ mutations. However, the introduction of the strong promoter P*_tet_* increased IS5 insertions by more than twofold. Similarly, the overexpression of the *ins5CB* operon at the *nmpC* locus led to a twofold increase in IS5 transposition upstream of the *fucAO* operon in comparison to the wildtype strain (Figure 9C). Based on these results, we conclude that (i) transcription (if any) from the downstream genomic regions (including P*_quuD_* and other operon promoters) does not traverse the *ins5CB* operon, most likely due to the presence one or more intrinsic terminator(s) and (ii) the overexpression of *ins5CB* significantly elevates IS5 transposition activity, suggesting that Ins5C and Ins5B, encoded by the *ins5CB* operon within IS5, play a role in stimulating IS5 transposition.

### 3.9. “Copy/Paste” Is the Dominant Mechanism for IS5 Transposition

There are multiple copies of IS5 per genome in some *E. coli* strains, suggesting that IS5 transposes primarily using a “Copy/Paste” mechanism. To confirm this, we deleted all eight active IS5 elements save for the IS5 copy at the *nmpC* locus. This strain (ZZ263) with a single active IS5 element was tested for Bgl^+^ mutations in comparison to the wildtype. As shown in Figure 10A, this IS5 element alone yielded approximately 60% of IS5-mediated Bgl^+^ mutations observed for the wildtype strain that carries nine active copies of IS5, consistent with the earlier conclusion that the *nmpC* copy of IS5 is the major contributor to overall IS5 transposition. Using colony PCR, 60 independent Bgl^+^ mutants were examined for the presence or the absence of the original IS5 copy at *nmpC*; the results are listed in Figure 10B. As can be seen, all the test mutants still possessed the native IS5 element at the *nmpC* locus. For each of these examined Bgl^+^ mutants, those two copies of IS5 (located within *lomR* and in the *cobU*/*yoeG* intergenic region, respectively) harboring point mutations were still unchanged. This was expected, since they are inactive elements. These results confirm that IS5 employs “Copy/Paste” as the primary transposition mechanism, although a low level of the “Cut/Paste” mechanism cannot be ruled out.

## 4. Discussion

A typical IS element transposition process simply involves the production of the transposase, formation of the transpososome, followed by the integration of the element into the target site, with the former two steps being relevant to the native loci of the transposing elements. However, despite extensive efforts devoted to the latter process (IS insertion into the target sites), little information is available as to how IS elements’ native loci (and their surrounding genomic regions) affect their transposition activities. Here, we show that IS5 elements distributed at different loci on the chromosome of *E. coli* strain BW25113 play distinct roles in overall IS5 transposition, although most elements are identical in their sequences. Out of 11 copies, the IS5 element embedded within the *nmpC* gene is the key contributor to IS5 transposition, while two mutant copies lost their hopping ability. IS5 transposition activity was heavily reliant on the promoter upstream of *nmpC*, with stronger promoter activities leading to more frequent transpositions. Unexpectedly, this *nmpC* copy of IS5 is the only element that transcribes its transposase gene *ins5A* at a detectable level. The transposase is most likely biased to transpose the IS5 element that encodes it. In addition, our results reveal that Ins5C and Ins5B exert a stimulatory role in IS5 transposition. These results not only fill in the gap in our knowledge on how an IS element’s native locus and its associated genomic contexts impact transposition but also contribute to our understanding of a complex but orchestrated process for transposition involving at least two genomic loci.

We demonstrate that the IS5 element located within the *nmpC* gene is the primary player in overall IS5 transposition. This conclusion was drawn through three independent assays, including the measurement of the Bgl^+^ mutations, the PPD^+^ mutations, and the SWM^+^ mutations, each of which showed a ~threefold decrease in IS5 insertion frequencies when the IS5 element was deleted from the *nmpC* locus. Meanwhile, in addition to the *nmpC* copy, another eight active copies of IS5 (with no point mutations) were examined for their roles in transposition as well, and their effects turned out to be much less pronounced. These results highlight the distinct roles of various copies of IS5 in overall transposition, which cannot be explained by these identical elements themselves. Each copy of IS5 is surrounded by different host genomic DNA that most likely affect the element’s contribution to transposition. Out of 11 IS5 copies, two IS5 elements, located at the *lomR* and *cobU*/*yoeG* loci (Table 1), seem to lose their transposition capability (or display an extremely low capability), since all identified new transpositions were not derived from these elements. These two IS5 copies carry multiple point mutations, leading to the alterations of one or more residues within three IS5-encoded proteins (Ins5A, Ins5B, and Ins5C), which likely account for their failure to transpose. Another possible reason could involve nearby genomic regions that might hinder the initial process of transposition, such as transpososome assembly [40,53].

IS5 carries an endogenous promoter-like (P*_ins5A_*) region upstream of the transposase gene *ins5A*, partially overlapping its IRL, but it is extremely weak, thus hardly transcribing *ins5A* [40,44]. This fact suggests that most transposases used for IS5 transposition are not from P*_ins5A_*-driven transcription. Instead, most of the transcription may be derived from upstream genomic regions near the inserted IS5 elements, since the *ins5A* genes from at least four IS5 elements are oriented in the same direction as their target genes/operons (Table 1). Indeed, this is the case. Using 5′ RACE, a clear cDNA product containing part of the *ins5A* gene was detected, and the corresponding mRNA was transcribed from the interrupted *nmpC* gene and the IS5 element nested within it. However, no detectable *ins5A* mRNA was obtained from other 10 IS5 elements under our experimental conditions. Meanwhile, as expected for an IS5 deletion, the deletion of the *nmpC* promoter led to a significant drop in transposition frequency, which was proportional to the strength of the promoter driving *nmpC* expression. These results, consistent with the observation of IS5 at *nmpC* being the most significant contributor to transposition, clearly demonstrate that the *nmpC* promoter is the main driver, efficiently transcribing *ins5A* and thus providing a sufficient transposase level for transposition. Of note, the *nmpC* gene encodes an outer membrane porin [54]. In most cases, this gene is inactivated by IS5 insertion, but such mutations do not cause any consequences to cell growth or physiology due to the presence of three other genes coding for porins that serve similar functions [51]. Most likely, the *nmpC* gene is not native to *E. coli*, since it is located on a defective lambdoid prophage on the chromosome. Instead, it may have been introduced into the genome when a lambda phage integrated into the chromosome [51]. As a foreign gene, *nmpC* itself appears to not be useful to *E. coli* cells. It may even be harmful, as its expression elevates the cellular sensitivity to colicins and phages [55], but the IS5 element it carries has important implications in increasing genome plasticity, for example, by activating or elevating proper genes/operons to adapt to environmental changes.

Our work showed that overexpression of *ins5A* at the *intS* locus did not significantly promote overall IS5 transposition, suggesting that the transposase proteins made from *ins5A* at this locus did not efficiently mediate the transposition of IS5 elements distal from this locus. Using a strain that was deleted for all active IS5 copies, save for IS5 at *nmpC*, we showed that this single copy of IS5 accounted for 60% of new transpositions observed for the wildtype strain that carry nine copies of active IS5 elements, suggesting that when using wildtype cells, most of the new transpositions originate from the IS5 element located at the *nmpC* locus. Based on these results, we postulate that the IS5 element at *nmpC* not only provides most of the transposases vital for transposition, but it also shows that the element itself is the most transposable element. In the latter case, the transposases preferentially transpose the IS5 element buried inside the *nmpC* gene. In other words, the IS5 transposase mainly functions in cis, biased towards the elements in which the enzyme is expressed.

The *ins5A* mRNA has been reported to be unstable [22]. Like several transposases encoded by other insertion sequences, IS5 transposase proteins could be readily lysed by host proteases before being transferred to other genomic regions [56,57]. Considering that frequent transpositions potentially might compromise genome integrity, preferential cis activity would be expected to be beneficial to the host, since it minimizes the activation of one element by others on the same genome.

It is worthy to note that we tried to directly confirm that most new transpositions are derived from the *nmpC* copy of IS5 by tagging this element (an A to T substitution at position −43 with respect to the 3′ end of IS5) but failed because the tagged IS5 element could no longer jump. Therefore, a new approach, such as moving the transposase gene and the IS5 element at various distances or identifying another tag without affecting transposition, is needed to directly determine if and how some copies of IS5 hop more frequently than others. Meanwhile, the observation that IS5 transposases mainly exert a cis effect helps to explain why those two mutant copies of IS5 fail to transpose. This could be due to the sum of two reasons, namely that (i) the altered transposases encoded within these mutant elements might lose their transposition capability and (ii) the transposases, synthesized from other IS5 elements, may not be able to transpose these elements.

In addition to the transposase (Ins5A), IS5 encodes another two small proteins, Ins5B and Ins5C, of unknown functions [22,58]. These two genes form an operon, divergently transcribed from *ins5A*, but transcription from its own promoter is negligible [44]. In this work, we found that blocking the possible transcription(s) from downstream regions of *nmpC*, including the *euuD* gene and other operons (all transcribed in the same direction as *ins5CB*), only slightly affected IS5 transposition. However, transcription initiated from these downstream promoters may not have affected the *ins5CB* operon because of the presence of intrinsic termination sequences. Nevertheless, this observation cannot rule out the possible effect of nearby genomic regions on *ins5CB*, such as the 3′ region of *nmpC*, whose role has yet to be examined. Furthermore, the *ins5CB* operon in several IS5 elements lies in the same direction as their target genes (Table 1), and they would conceivably be transcribed together with its target gene/operon simultaneously. In this case, genomic contexts around IS5 elements affect expression not only of the transposase gene but also of the *ins5CB* operon.

To further explore the possible function of external promoters, a constitutive promoter was added immediately downstream of *nmpC*, directly driving the *ins5CB* operon. A significantly elevated IS5′s transposition activity was observed, presumably by increasing the expression of *ins5CB*. This suggests that these two proteins indeed play a role in IS5 transposition. Like IS5, many other IS elements encode one or two other proteins in addition to the transposase protein. In several cases, these proteins are short or truncated derivatives of the transposases, encoded by only part of the transposase gene (usually the 5′ region). These accessory proteins may interfere with the transposition activity by binding to the promoter of the transposase gene, thus impeding its expression or by competing with the transposase to bind to the IRL [59,60,61,62], thus impairing the transposition activity. In the case of IS5, Ins5B and Ins5C apparently are not inhibitory to IS5 transposase expression or its transposition activity, as their presence apparently stimulates transposition. An early study indicated that the IS5 transposase mRNA is unstable [22]. Several other reports showed the presence of transcription terminator-like sequences in the IS5 element, resulting in premature transcriptional termination [63,64]. Ins5B and Ins5C might function to stabilize *ins5A* mRNA or to assure its complete transcription. Alternatively, these small proteins might aid in the formation or stabilization of the transposase complexes or assemble the functional transpososomes. It is unknown if Ins5B and/or Ins5C can promote the transposition of other IS5 elements present in the same genome. Clearly, more studies are needed to examine the actual functions of Ins5B and Ins5C in IS5 transposition.

As shown in Figure 6D, *nmpC* expression is under the control of multiple regulatory proteins and of the small RNA RyhB as well. Among these regulators, OmpR functions in response to external osmolarity and pH; Crp activity depends on cAMP levels, while RyhB expression is heavily affected by the iron availability. As cited earlier, IS5 insertion into its hotspot is DNA structure-dependent, is induced by stress environmental conditions (such as starvation), and is regulated by DNA-binding proteins. All these factors, involving transcriptional/post-transcriptional regulators, DNA-structuring proteins, environmental conditions, and DNA structures, will merit further detailed investigation as to their impacts on the transposing IS5 element(s) located at the *nmpC* locus or other native loci, especially with respect to their effects on the expressions of both *ins5A* and the *ins5CB* operon and on the assembly of functional transpososomes.

## Figures and Tables

**Figure 1 microorganisms-12-02600-f001:**
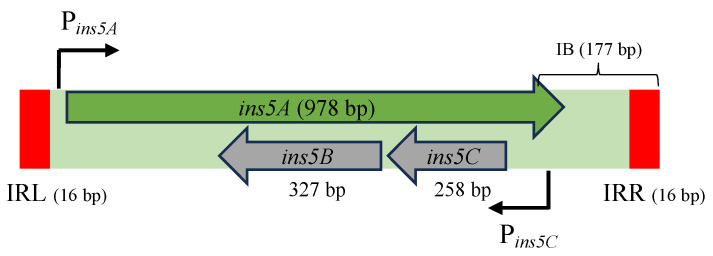
A schematic diagram of the IS5 element. IS5 is 1195 bp long and is flanked by two 16 bp terminal inverted repeat (IR) sequences. IS5 carries three overlapping genes, among which *ins5A* encodes the transposase essential for transposition, while the divergently transcribed *ins5CB* operon encodes two small proteins with a previously unknown function. P*_ins5A_* and P*_ins5C_* refer to the *ins5A* promoter and the *ins5CB* promoter, respectively. Located in the 3′ end is a 177 bp IB (internal bent) region harboring A-tracts and a cortically located IHF binding site. IRL and IRR denote the inverted repeat sequence on the left end and on the right end, respectively.

**Figure 2 microorganisms-12-02600-f002:**
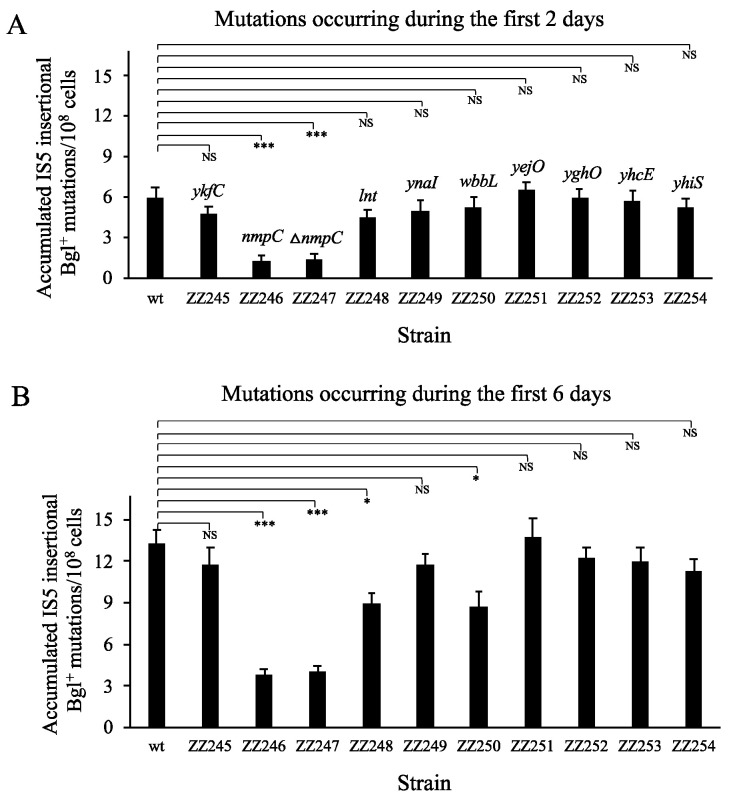
Impacts of deleting active IS5 elements on Bgl^+^ mutations. (**A**) Effects of deleting each active IS5 copy on Bgl^+^ mutations occurring during the first two days (n = 15). (**B**) Effects of deleting each IS5 copy on Bgl^+^ mutations occurring during the first six days (n = 15). Standard Bgl^+^ mutation assays were conducted on salicin M9 minimal agar plates. During the incubation at 30 °C, visible colonies (representing Bgl^+^ mutations) were counted daily, beginning on day 2 post-inoculation. The colonies arising on day 2 were derived from pre-existing mutations and they were subtracted from the total Bgl^+^ mutants. Colony PCR was employed to distinguish IS5 insertional mutants from IS1 insertional mutants and other non-insertional mutants. The mutation frequencies were normalized to the accumulated Bgl^+^ mutations per 10^8^ cells during a given incubation period. Data are plotted as the mean ± SD (two-sample *t*-tests between wt and each strain individually). NS denotes no significance and indicates a *p*-value ≥ 0.05; * indicates a *p*-value < 0.05 and *** indicates a *p*-value < 0.001.

**Figure 3 microorganisms-12-02600-f003:**
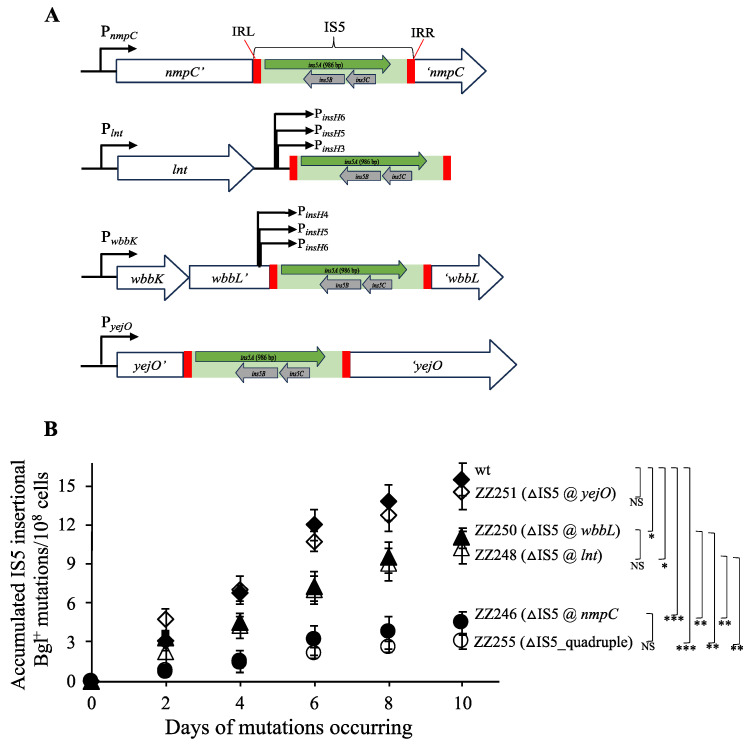
Impacts of four copies of IS5 harboring upstream genomic promoters on Bgl^+^ mutations. (**A**) Schematic diagrams showing the presence of promoter regions upstream of four IS5 elements. At the *nmpC* locus, the *nmpC* promoter P*_nmpC_* drives the interrupted *nmpC* and the *ins5A* transposase gene in the IS5 element. At the *lnt* locus, in addition to the *lnt* promoter, P*_lnt_*, three other putative promoters (P*_insH3_*, P*_insH5_*, and P*_insH6_*) with unknown functions are present in the *lnt*/IS5 intergenic region. These four promoters might drive *ins5A* transcription. At the *wbbL* locus, the *wbbK* promoter (P*_wbbK_*) and three other putative promoters (P*_insH4_*, P*_insH5_*, and P*_insH6_*) of unknown function are present upstream of IS5. At the *yejO* locus, the uncharacterized *yejO* promoter, P*_yejO_*, might drive transcription of both the interrupted *yejO* gene and the inserted IS5 element. (**B**) Bgl^+^ mutations over time comparing five IS5 deletion mutants with the wildtype (n = 15). Among those five test mutants, the IS5 element(s) was/were deleted individually at *nmpC*, *lnt, wbb*L, and *yejO* or at all of these four loci. Data are plotted as the mean ± SD (one-way ANOVA with Tukey–Kramer’s post hoc test). Mutation frequencies at day 10 were subject to statistical analyses. NS denotes no significance and indicates a *p*-value ≥ 0.05; * indicates a *p*-value < 0.05; ** indicates a *p*-value < 0.01; *** indicates a *p*-value < 0.001.

**Figure 4 microorganisms-12-02600-f004:**
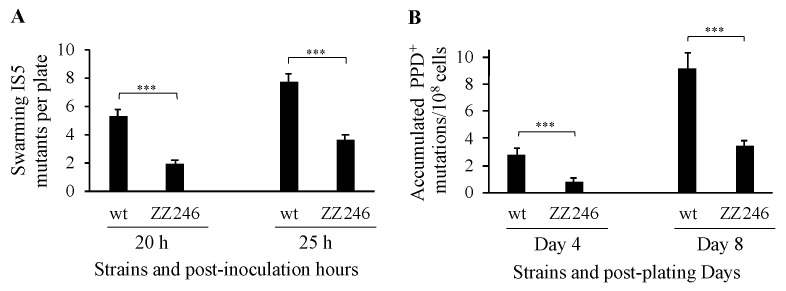
The IS5 element at *nmpC* is important for IS5-mediated swarming mutations (SWM^+^) and propanediol growth mutations (PPD^+^). (**A**) The effect of deleting IS5 at *nmpC* on IS5 insertional SWM^+^ mutations (n = 12). SWM^+^ mutations were conducted on LB semi-agar plates. SWM^+^ mutants were counted after 20 h and 25 h incubations, and the IS5 insertional SWM^+^ mutants were distinguished by PCR. (**B**) The effect of deleting IS5 at *nmpC* on PPD^+^ mutations (n = 15). The PPD^+^ mutation assays were conducted on propanediol agar plates. The PPD^+^ colonies (representing IS5 insertional mutations) were counted daily after plating. The mutation frequencies were normalized to mutations per 10^8^ cells during a given period of incubation. Data are plotted as the mean ± SD (two-sample *t*-test). *** indicates a *p*-value < 0.001.

**Figure 5 microorganisms-12-02600-f005:**
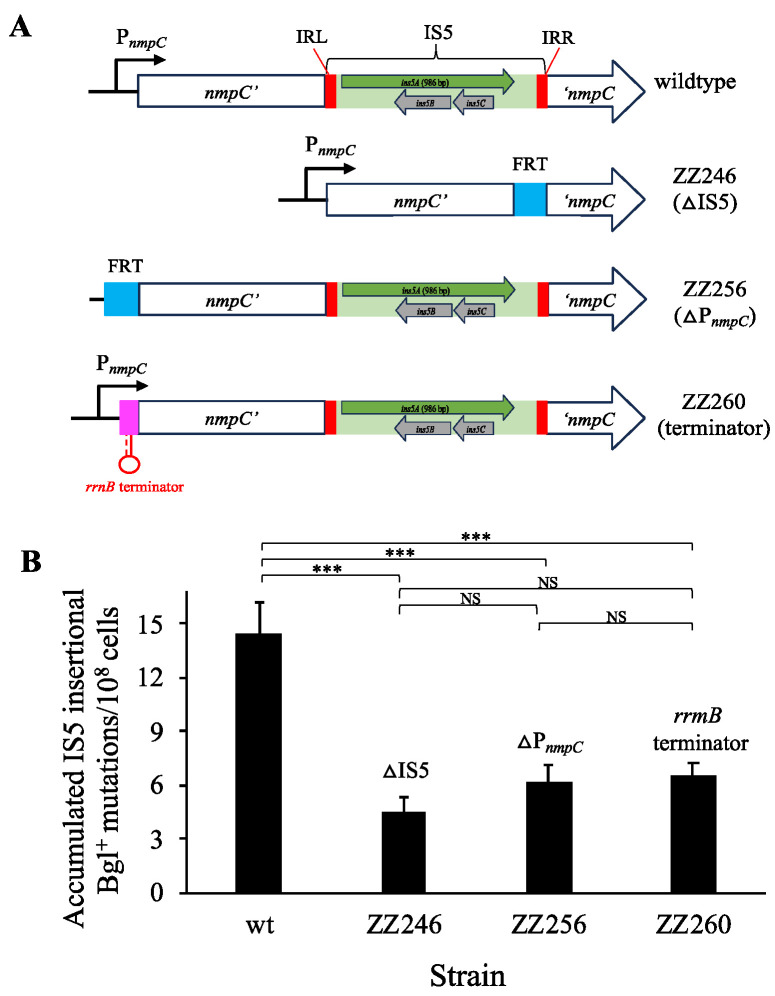
The *nmpC* promoter (P*_nmpC_*) is critical for IS5-mediated Bgl^+^ mutations. (**A**) Schematic diagrams showing (i) P*_nmpC_* driving *nmpC* and the IS5 transposase gene *ins5A* in the wildtype; (ii) the IS5 deletion at *nmpC* (strain ZZ246); (iii) the deletion of P*_nmpC_* in strain ZZ256; and (iv) the insertion of an *rrnB* terminator between P*_nmpC_* and *nmpC* in strain ZZ260. (**B**) Effects of deleting P*_nmpC_* or blocking P*_nmpC_* on IS5 insertion upstream of the *bglGFB* operon (n = 15). The figure shows the accumulated IS5 insertional Bgl^+^ mutations arising during a 6-day incubation. Data are plotted as the mean ± SD (one-way ANOVA with Tukey–Kramer’s post hoc test). NS denotes no significance and indicates a *p*-value ≥ 0.05; *** indicates a *p*-value < 0.001.

**Figure 6 microorganisms-12-02600-f006:**
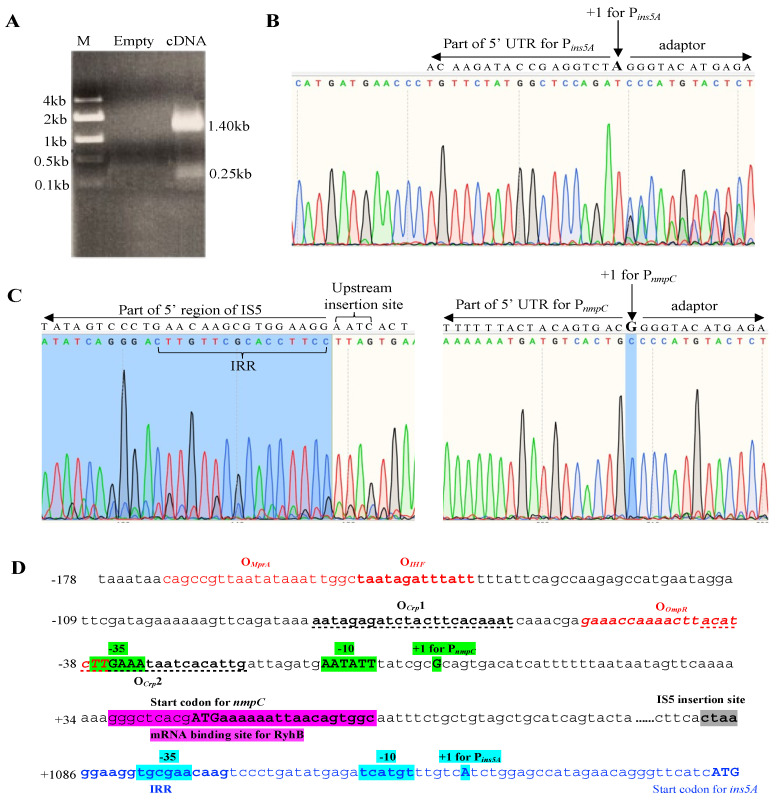
Examining IS5 elements that transcribe detectable amounts of *ins5A* mRNA using 5′ RACE. The wildtype strain BW25113 was cultured in glycerol M9 minimal medium. Total RNA preparation, rRNA removal, mRNA purification, cDNA synthesis, and amplification with an IS5-specific oligo (binding to the *ins5A* gene near its start codon) are described in Materials and Methods. (**A**) Agarose gel picture showing two cDNA products amplified using the IS5 specific oligo. The 0.25 kb product is part of the *ins5A* cDNA, while the 1.4 bp is part of *nmpC*/IS5 fusion cDNA. (**B**) A sequencing chromatogram of the *ins5A* cDNA (0.25 kb) revealing the TSS. The capital “A” is the TSS determined for P*_ins5A_* by 5′ RACE. The sequence on the left side of the TSS is the beginning sequence of the 5′ untranslated region (UTR) of *ins5A*, while the sequence on the right side is derived for a sequencing adaptor provided within the Takara Bio kit. (**C**) Sequencing chromatograms of the IS5/*nmpC* cDNA (1.4 kb). The left panel shows the 5′ end region of IS5 (including the IRR) and the IS5 insertion site CTAA. On the right panel, the capital “G” is the TSS determined for P*_nmpC_*, which is the same TSS as reported for *nmpC* [51,52]. Flanking the TSS are part of 5′ UTR for P*_nmpC_* (left) and part of a sequencing adaptor provided by Takara (right). (**D**) DNA sequences of P*_nmpC_* and P*_ins5A_* with annotations. Within the *nmpC* regulatory region (black), the TSS (+1), −10 element, and −35 element are highlighted in green. Two Crp binding sites are bolded and located above the dotted lines. The binding sites for OmpR, IHF, and MprA are in red. The region interacting with the RyhB (small RNA) is highlighted in purple. The majority of *nmpC*, between the IS5 insertion site (ctaa) and its beginning region, is represented by six dots (……). For P*_ins5A_* (blue), the TSS (+1), −10 element (putative), and −35 element (putative) are highlighted in cyan.

**Figure 7 microorganisms-12-02600-f007:**
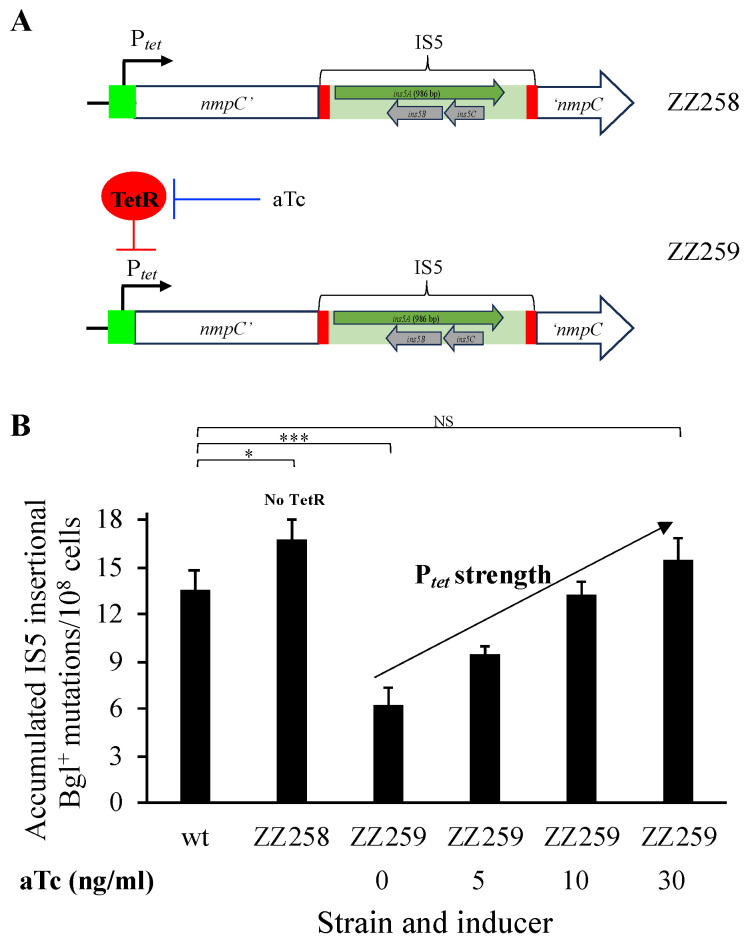
Greater promoter strength leads to increased IS5 transposition. (**A**) A diagram showing P*_tet_* substitution for P*_nmpC_* and its regulation by TetR. In the absence of the repressor TetR (ZZ258), P*_tet_* displays its maximal activity. When TetR is present (ZZ259), P*_tet_* is repressed, and the repression is released by the inducer anhydrotetracycline (aTc), with more aTc leading to greater promoter activity. (**B**) Effects of titrating P*_tet_* on Bgl^+^ mutations (n = 15). Bgl^+^ mutation assays were performed under the standard conditions (see Figure 2 and Section 2). In strain ZZ258, P*_tet_* drives *nmpC* and the *ins5A* gene in IS5 in the absence of TetR. Strain ZZ259 is same as ZZ258 except that it constitutively produces TetR. For the latter strain, the P*_tet_* strength can be titrated using aTc at 0 to 30 ng/mL. Data are plotted as the mean ± SD (two-sample *t*-test between wt and each strain individually). NS denotes no significance and indicates a *p*-value ≥ 0.05; * indicates a *p*-value < 0.05; *** indicates a *p*-value < 0.001.

**Figure 8 microorganisms-12-02600-f008:**
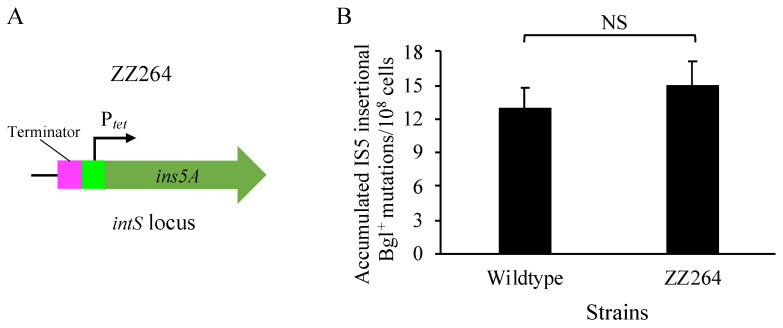
*ins5A* expression at the *intS* locus does not promote IS5 transposition. (**A**) A schematic diagram showing P*_tet_* driving *ins5A* transcription at the *intS* locus. (**B**) Effects of the overexpression of *ins5A* at the *intS* locus on Bgl^+^ mutations (n = 15). Standard Bgl^+^ mutation assays were carried out. The figure shows the frequencies of IS5 insertional mutations arising during the first 6 days of incubation. Data are plotted as the mean ± SD (two-sample *t*-test). NS denotes no significance.

**Figure 9 microorganisms-12-02600-f009:**
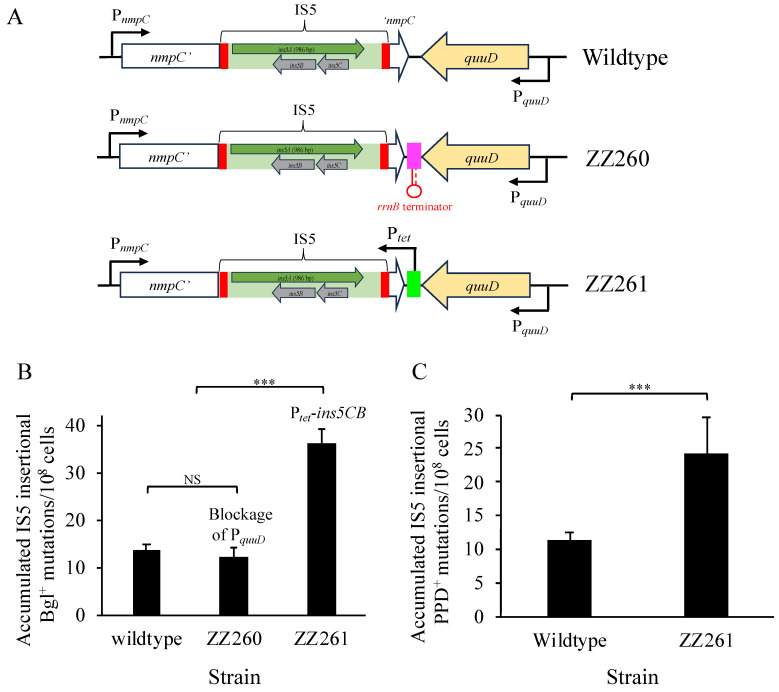
Increased *ins5CB* expression enhances IS5 transposition. (**A**) Schematic diagrams showing IS5′s downstream genomic region at *nmpC* and its alterations. As shown on the top of the diagram, the *quuD* gene is transcribed in the same direction as the *ins5CB* operon, and its promoter (P*_quuD_*) might impact *ins5CB* transcription. To block P*_quuD_*, an *rrnB* terminator was inserted into the *nmpC*/*quuD* intergenic region (middle part of the diagram). To increase the transcription of *ins5CB*, a P*_tet_* promoter was added downstream of *nmpC*, driving the *inc5CB* operon (bottom of the diagram). (**B**) Effects of blocking P*_quuD_* or increasing *ins5CB* transcription on Bgl^+^ mutations (n = 15). Strains ZZ260 (P*_quuD_* blockage) and ZZ261 (P*_tet_* driving *ins5CB*) were assayed for Bgl^+^ mutations in comparison to the wildtype. The figure shows the frequencies of IS5 insertional mutations that occurred during the first 6 days post-plating. (**C**) Effects of the overexpression of *ins5CB* on PPD^+^ mutations. Standard PPD^+^ mutation assays were carried out. The figure shows the frequencies of IS5 insertional mutations arising during the first 10 days of incubation. Data are plotted as the mean ± SD (one-way ANOVA with Tukey–Kramer’s post hoc test (**B**); two-sample *t*-test (**C**)). NS denotes no significance; *** indicates a *p*-value < 0.001.

**Figure 10 microorganisms-12-02600-f010:**
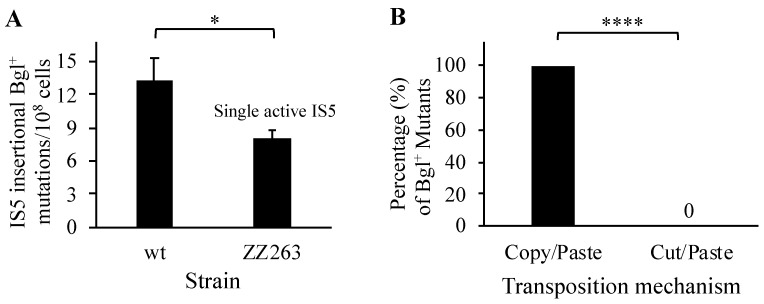
IS5 transposes using the “Copy/Paste” mechanism. (**A**) The single IS5 element at *nmpC* can transpose (n = 15). Strain ZZ263 (deleted for all active IS5 copies except for the one at *nmpC*) was used for Bgl^+^ mutation assays. (**B**) IS5 uses “Copy/Paste” as its primary transposition mechanism (n = 60). Twenty independent IS5 insertional Bgl^+^ mutants from each of three separated for Bgl^+^ mutation assays using strain ZZ263 were examined for the presence or absence of the original IS5 at the *nmpC* locus. For a given mutant, the presence of the original IS5 copy at *nmpC* denotes a “Copy/Paste” transposition, while the absence of the *nmpC* copy denotes a “Cut/Paste” transposition. Data are plotted as the mean ± SD (two-sample *t*-test). * indicates a *p*-value < 0.05; **** indicates a *p*-value < 0.0001.

**Table 1 microorganisms-12-02600-t001:** Locations, directions and similarity to wildtype IS5 for 11 copies of IS5 found on BW25113 chromosome.

Copy	Coordinates * (Kb)	Target Genes **	Orientation ***	Similarity to IS5
1	269.8/270.8	*ykfC*	Reverse	100%
2	570.2/571.2	*nmpC*	Direct	100%
3	683.5/684.5	*gltI*/*lnt*	Direct	100%
4	1390.3/1391.3	*ynaI*/*ynaJ*	Reverse	100%
5	1422.0/1423.0	*lomR*	Reverse	91.5%
6	2059.8/2060.8	*cobU*/*yoeG*	Reverse	99.6%
7	2095.4/2096.4	*wbbL*	Direct	100%
8	2282.5/2283.6	*yejO*	Direct	100%
9	3123.5/3124.5	*yghO*	Reverse	100%
10	3359.1/3360.1	*yhcE*	Reverse	100%
11	3635.4/3646.6	*yhiS*	Reverse	100%

* Denotes the locations of IS5 on the chromosome of wildtype strain BW25113. ** For the targets with one gene, IS5 is inserted inside the gene. For the targets with two genes, IS5 is inserted into the intergenic region. *ynaI* and *ynaJ* are divergently transcribed, while *cobU* and *yoeG* are transcribed in the same direction. *** “Reverse” or “Direct” denote that the *ins5A* gene and its target gene are oriented in the opposite direction or in the same direction, respectively.

## Data Availability

The original contributions presented in this study are included in the article/Appendix A. Further inquiries can be directed to the corresponding authors.

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
