# Peer review of "Investigating How Genomic Contexts Impact IS5 Transposition Within the Escherichia coli Genome"

_microorganisms, 2024, doi:10.3390/microorganisms12122600_

Round 1

Reviewer 1 Report

Comments and Suggestions for Authors

Comments to the manuscript Microorganisms-3344478 titled “Effects of Genomic Contexts on IS5 Transposition in Escherichia coli”

The authors analyzed the influence of genomic context on the transposition of the IS5 element in an E. coli strain BW25113 carrying 11 copies of IS on chromosome. In addition, during the analysis, they found that two copies carrying mutations in the transposable gene were unable to transpose. The authors designed and constructed the mutants well (without individual IS5 elements), and chose methods that demonstrated both the overall activity of the IS5 copy located in the nmpC operon and the contribution of the PnmpC gene promoter to transposition as well as the mechanism of transposition. The conclusions reached are supported by the results obtained, although they could be even stronger. It would be possible to work more on the gene regulation of transcription of nmpC gene, since to a large extent the expression of the transposase and the transposition of the IS5 itself depend on it in order to conclude whether the insertion into the nmpC gene contributes to adaptability. In Pseudomonas, it is known that the expression of the transposase is regulated not only by the genomic context but also by a large number of global cellular regulators. (Novovic KD, Malesevic MJ, Filipic BV, Mirkovic NL, Miljkovic MS, Kojic MO, Jovčić BU. PsrA Regulator Connects Cell Physiology and Class 1 Integron Integrase Gene Expression Through the Regulation of lexA Gene Expression in Pseudomonas spp. Curr Microbiol. 2019 Mar;76(3):320-328. doi: 10.1007/s00284-019-01626-7. Epub 2019 Jan 25. PMID: 30684026.). I think it would be more appropriate if the authors used the term "located" instead of "situated". I'm not sure if it's a typo or if the authors really used sodium sulfate (60 g of Na2SO4; Line 130) instead of sodium phosphate to make M9 medium. If it was used intentionally, a reason should be given why.

Line 239, Phrase “RNase inactivation “ is repeated twice

I find the writing style a bit strange, so it could be worked on a little. There are several errors in writing degrees Celsius,

Comments on the Quality of English Language

I am not qualified to grade English.

Author Response

Comment 1: It would be possible to work more on the gene regulation of transcription of nmpC gene, since to a large extent the expression of the transposase and the transposition of the IS5 itself depend on it in order to conclude whether the insertion into the nmpC gene contributes to adaptability. In Pseudomonas, it is known that the expression of the transposase is regulated not only by the genomic context but also by a large number of global cellular regulators. (Novovic KD, Malesevic MJ, Filipic BV, Mirkovic NL, Miljkovic MS, Kojic MO, Jovčić BU. PsrA Regulator Connects Cell Physiology and Class 1 Integron Integrase Gene Expression Through the Regulation of lexA Gene Expression in Pseudomonas spp. Curr Microbiol. 2019 Mar;76(3):320-328. doi: 10.1007/s00284-019-01626-7. Epub 2019 Jan 25. PMID: 30684026.).  

Response 1: We appreciate your positive comments to our work! Thanks for introducing this reference to us! We read this paper authored by Novovic et al with great interest and feel it may not be relevant to our manuscript. First, our work focuses on studying genomic context effects on the transposition of an insertion sequence element, IS5 (a small transposon capable of independent transposition from one location to another). We found that the IS5 copy located within the nmpC gene is the most important element for overall IS5 transposition. This effect is attributed to the nmpC promoter (PnmpC). Second, we are aware that several regulatory proteins (and one small RNA) regulate PnmpC (Figure 6D), thereby proposing one major future study in the Discussion section (last paragraph) to characterize how these regulators would impact IS5 transposition by exerting their effects on PnmpC and thus varying the IS5 transposase amounts.  

However, the Novovic et al paper describes how several host regulatory proteins directly or indirectly regulate expression of the integrase gene within an integron. Based on our understanding, this paper doesn’t explore how the genomic DNAs affect the integrase gene transcription. In addition, an integron is very different from an IS element as to their structures and mobile mechanisms (PMID: 24847022). Typically, each IS element carries a transposase gene that is flanked by two short IR sequences, encoding the transposase protein that enables independent transposition of the element from one location to another on the chromosome. However, an integron consists of three parts: an integrase gene intI (encoding an integrase), a recombination site attI and an integron-associated promoter Pc. The integrons usually cannot transpose by themselves and instead they mainly act to acquire and express exogenous DNA fragments. 

Based on these considerations, we feel it is not appropriate to cite this Novovic et al paper as a background for our study on the impact of genomic contexts on transposition of an IS element.   Meanwhile, we made several modifications to the Abstract and the Introductory section in order to make them more readable and understandable. 

Comment 2: I think it would be more appropriate if the authors used the term "located" instead of "situated".  

Response 2: As suggested, we have now changed “situated” on Lines 67, 273 and 622 to “located” or “distributed”. 

Comment 3: I'm not sure if it's a typo or if the authors really used sodium sulfate (60 g of Na2SO4; Line 130) instead of sodium phosphate to make M9 medium. If it was used intentionally, a reason should be given why. 

Response 3: Thanks for pointing out this typo! We now changed “Na2SO4” to “Na2HPO4” on Line 131. 

Comment 4: Line 239, Phrase “RNase inactivation” is repeated twice. 

Response 4: Thanks for finding this error! The second “RNase inactivation” phase has been deleted (see Line 240). 

Comment 5: I find the writing style a bit strange, so it could be worked on a little. There are several errors in writing degrees Celsius. 

Response 5: As mentioned above, some modifications were made on both Abstract and Introduction sections. For the degree Celsius, we now use the standard Celsius symbol throughout the manuscript! 

Reviewer 2 Report

Comments and Suggestions for Authors

They demonstrated that the IS5 element at the nmpC locus is responsible for most IS5 transposing among the 11 copies of IS5 in the BW25113 strain, and there is little direct and/or synergistic effects by other copies. 

They identified the real promoter of the IS5 transposase gene by isolating its transcript and its product prefers to transpose the IS5 copy in cis.  

They suggested the two gene products encoded divergently in the IS5 transposase gene may have positive functions.

They also indicate the IS5 transposition occurs in copy/paste rather than cut/paste.

These conclusions are deduced from the results of well-designed experiments in molecular genetics, and thus I recommend this excellent manuscript to be published in the present form. 

Author Response

Thanks for your very positive comments to our work! We are gold that you like our paper. Since there are no criticisms or suggestions provided, we did not make any modifications based on Reviewer 2's review.